# Transcriptional Analysis of the Differences between ToLCNDV-India and ToLCNDV-ES Leading to Contrary Symptom Development in Cucumber

**DOI:** 10.3390/ijms24032181

**Published:** 2023-01-22

**Authors:** Thuy T. B. Vo, Won Kyong Cho, Yeonhwa Jo, Aamir Lal, Bupi Nattanong, Muhammad Amir Qureshi, Marjia Tabssum, Elisa Troiano, Giuseppe Parrella, Eui-Joon Kil, Taek-Kyun Lee, Sukchan Lee

**Affiliations:** 1Department of Integrative Biotechnology, Sungkyunkwan University, Suwon 16419, Republic of Korea; 2College of Biotechnology and Bioengineering, Sungkyunkwan University, Suwon 16419, Republic of Korea; 3Institute for Sustainable Plant Protection of the National Research Council (IPSP-CNR), 80055 Portici, Italy; 4Department of Plant Medicals, Andong National University, Andong 36729, Republic of Korea; 5Risk Assessment Research Center, Korea Institute of Ocean Science & Technology, Geoje 53201, Republic of Korea

**Keywords:** cucumber, symptom development, RNA-seq, tomato leaf curl New Delhi virus, infectivity comparison

## Abstract

Tomato leaf curl New Delhi virus-ES (ToLCNDV-ES), a high threat to cucurbits in the Mediterranean Basin, is listed as a different strain from the Asian ToLCNDV isolates. In this study, the infectivity of two clones previously isolated from Italy and Pakistan were compared in cucumbers, which resulted in the opposite symptom appearance. The swapping subgenome was processed; however, the mechanisms related to the disease phenotype remain unclear. To identify the disease-associated genes that could contribute to symptom development under the two ToLCNDV infections, the transcriptomes of ToLCNDV-infected and mock-inoculated cucumber plants were compared 21 days postinoculation. The number of differentially expressed genes in ToLCNDV-India-infected plants was 10 times higher than in ToLCNDV-ES-infected samples. The gene ontology (GO) and pathway enrichment were analyzed using the Cucurbits Genomics Database. The flavonoid pathway-related genes were upregulated in ToLCNDV-ES, but some were downregulated in ToLCNDV-India infection, suggesting their role in resistance to the two ToLCNDV infections. The relative expression levels of the selected candidate genes were validated by qRT-PCR under two ToLCNDV-infected conditions. Our results reveal the different infectivity of the two ToLCNDVs in cucumber and also provide primary information based on RNA-seq for further analysis related to different ToLCNDV infections.

## 1. Introduction

Viruses that infect plants can be divided into DNA and RNA viruses based on their genome type. Of the known DNA viruses that infect plants, geminiviruses are one of the largest plant viral families. They include more than 520 species, which are further divided into 14 genera according to their genome structures, insect vectors, and host ranges [1]. In general, geminiviruses are composed of circular single-stranded (ss) DNA genomes ranging 2.5–5.2 kb in length, which are packaged in icosahedral-twinned particles [2,3]. Geminiviruses infect a wide range of plant hosts, such as monocotyledonous and dicotyledonous crops, causing severe economic damage to their production worldwide.

*Tomato leaf curl New Delhi virus* (ToLCNDV) is a member of the genus *Begomovirus* of the family *Geminiviridae* [4]. In nature, ToLCNDV is usually transmitted by whiteflies (*Bemisia tabaci*); however, some published reports have demonstrated that ToLCNDV might be transmitted by mechanical inoculation or seeds [5,6,7]. This virus has a wide host range, including *Solanaceae* and *Cucurbitaceae* [8,9]. In addition, infection with ToLCNDV has been reported in other plants belonging to the *Papaveraceae, Malvaceae, Fabaceae*, and *Acanhaceae* [10,11,12,13]. Previously, ToLCNDV was identified mostly in South and Southeast Asian countries until 2012; however, the virus has spread to Southern Europe and Mediterranean countries with the new variant named ToLCNDV-ES [14,15,16,17,18,19]. Currently, ToLCNDV-ES is prevalent in Europe, resulting in substantial losses in cucurbit crop cultivation.

Viruses rely on host plants for their reproduction and metabolic processes. Numerous studies have revealed the molecular mechanisms and host factors involved in the interaction between viruses and host plants [20,21]. Transcriptome analyses have been used to identify host factors, and RNA-sequencing is widely used for expression profiling. RNA-sequencing (RNA-seq) is a developed approach to transcriptome profiling using high-throughput sequencing methods, which provide more precise measurements of transcript levels than other methods, such as Sanger sequencing and microarray-based methods [22,23]. Therefore, this technique is widely applied to evaluate the interaction between the virus and host plants and also offers a global view of gene expression changes in plants during viral infection. The differentially expressed genes (DEGs) caused by cucurbit chlorotic yellow virus (CCYV) infection in cucumber were analyzed to provide comprehensive transcriptomic information and improve the understanding of host–virus interactions [24]. Three major enrichment pathways, photosynthesis-antenna protein, phenylalanine metabolism, and phenylpropanoid biosynthesis, were identified under CCYV infection conditions. Recent reports have shown comparative transcriptome analysis between cucumber cultivars resistant and susceptible to cucumber mosaic virus [25], which can potentially be applied in breeding programs for virus-resistant crops. The expression of genes responsible for methylation, phosphorylation, cell wall organization, and carbohydrate metabolism prevailed in resistance “Heliana”, whereas the susceptible “Vanda” preferred expression for chromosome condensation and glucan biosynthesis. Many studies based on RNA-seq to clarify and identify resistance genes under ToLCNDV infection have recently been reported. In 2017, RNA-seq was employed to identify differentially regulated genes during infection with ToLCNDV isolated from India in resistant and susceptible potatoes [26]. Similarly, Spanish scientists recently compared transcript levels between mock- and ToLCNDV-ES-infected melon between the resistant WM-7 genotype and a PS-susceptible cultivar to identify candidate genes for resistance [27]. However, no studies have focused on the biological features of ToLCNDV in the same host.

Cucumber was used to investigate the interactions between host plants and ToLCNDV isolated from different geographical areas. This crop is an important cucurbit cultivated worldwide and is a major food source [28,29]. However, their production is limited, especially in Spain, under the threat of ToLCNDV-ES [30]. Thus, studying the response of cucumber to various pathogens is necessary to control disease and improve crop yield by screening sources resistant to ToLCNDV. In this study, infectious clones of ToLCNDV from the Indian subcontinent in Asia (Pakistan) and the Mediterranean Basin (Italy) were used to compare infectivity in cucumbers. The infected plants showed contrasting disease phenotypes related to various responses to the two ToLCNDV infections. The swapping subgenome between the two clones was processed to determine the effect of each viral DNA component on symptom induction. A transcriptome analysis of the infected cucumbers was conducted to understand the molecular mechanisms in the response of the host plant to infection with two isolates of ToLCNDV. These data may provide a better understanding of the molecular mechanism of different ToLCNDV–host interaction-related symptom development progress and serve as a basis for searching candidate sources to reduce the impact of ToLCNDV.

## 2. Results

### 2.1. Infectivity of Two ToLCNDV Clones in Cucumber

Two infectious clones, ToLCNDV-India, derived from Pakistan, and ToLCNDV-ES, from Italy, were inoculated into 3 week old cucumber plants, and leaf samples were collected at different time points to compare their infectivity (Figure 1A).

At 3 weeks postincubation (wpi), the mock-treated (pCAMBIA 1303 empty) cucumber did not show any visible disease symptoms. Infection with ToLCNDV-India caused severe leaf curling and mosaic yellowing in infected cucumber leaves. In contrast, cucumber infected with ToLCNDV-ES showed a normal phenotype (Figure 1B). The results observed in cucumber were different from those in *Nicotiana benthamiana* (Table 1).

To confirm the infection of two different ToLCNDV strains in cucumber plants, PCR amplification was performed using ToLCNDV-specific primers, as described in the Section 4. The results showed detectable amplicons from all cucumbers infected with two different ToLCNDV strains (Figure 1C). The relative viral titer by qPCR also showed the difference in the accumulation of the two ToLCNDV strains after inoculation (Figure 2). The DNA A segment of ToLCNDV-India showed high accumulation after 2 wpi, whereas DNA B showed increased accumulation at 3 wpi (Figure 2A,B). In comparison, titers of ToLCNDV-ES were much lower at 2 and 3 wpi. ToLCNDV-ES DNA A showed the highest accumulation at 6 wpi and DNA B at 8 wpi. The difference in titer may affect the disease phenotype appearance between the two ToLCNDV strains in cucumber.

### 2.2. Determination of the DNA Component Responsible for Viral Disease Symptoms in Cucumber 

To examine which DNA component may induce different viral symptoms between the two ToLCNDV strains, the subgenome of each isolate was swapped and challenged on the cucumber plants via agroinoculation (Figure 3). Two different combinations, ToLCNDV-India DNA A + ToLCNDV-ES DNA B (A_In_B_ES_) and ToLCNDV-ES DNA A + ToLCNDV-India DNA B (A_ES_B_In_), were used to infect the cucumber plants. The first combination of A_In_B_ES_ showed mild mosaic yellowing symptoms in the infected cucumber plants (Figure 4A, Table 1). The PCR showed a high level of ToLCNDV infection in the cucumber plants at one to three weeks postinfection (Figure 4B). Similarly, the second combination of A_ES_B_In_ also induced mild viral symptoms in the infected cucumber plants (Figure 4C, Table 1). According to the PCR results, all fragments of DNA A and DNA B were detected in the inoculated cucumber; however, the samples in 1 wpi showed most slightly bands for A_ES_B_In_ infection (Figure 4D). 

### 2.3. Identification of Differentially Expressed Genes

Twelve libraries were generated from three different groups (mock, ToLCNDV-India, and ToLCNDV-ES), which were further paired-end sequenced using the NovaSeq 6000 system. The number of sequenced reads ranged from 31 to 42 million (Table 2). The GC content ranged from 44.75% to 45.62%, and the Q20 percentage ranged from 97.42% to 98.98%. The mapped reads on the cucumber transcripts were subjected to the identification of DEGs using the DESeq2 package. Compared to the mock condition, the DEGs for the ToLCNDV-India and ToLCNDV-ES conditions, were considered significant with fold changes ≥ 2 and adjusted *p*-values ≤ 0.01. According to the results, the number of identified DEGs under the ToLCNDV-India infection compared to ToLCNDV-ES condition is shown via volcano plot (Figure 5A,B). In total, 180 genes were differentially regulated in the ToLCNDV-ES infection group and 1985 genes in the ToLCNDV-India infection group. Among them, 102 DEGs were identified under both conditions (Figure 5C). Only two of these DEGs (CsaV3_7G001540 and CsaV3_7G004500) showed opposite expression patterns between the two isolates. CsaV3_7G001540 encodes the expansin-like A2 protein, which plays an important role in cell wall modification during plant growth and elongation [31]; this DEG was induced under ToLCNDV-India and repressed under the ToLCNDV-ES infection group. Another gene, CsaV3_7G004500, had a similar expression tendency and encoded an unknown functional protein.

By comparing only upregulated genes in both conditions, 60 genes were upregulated in the ES infection group, whereas 1043 genes were upregulated in the India infection group (Figure 5D). The number of downregulated genes in the India infection group (942 DEGs) was 7.85 times higher than that in the ES infection group (120 DEGs) (Figure 5E). These results suggest that ToLCNDV-India infection had a greater impact on the transcriptomic reorganization than ToLCNDV-ES. The most upregulated gene in response to ToLCNDV-ES was subtilisin-like protease SBT1, together with the gene encoding the carboxyl methyltransferase protein (Appendix A). Subtilisin-like proteases are serine proteases, which are one of the best-characterized groups of proteolytic enzymes, and play key roles in a variety of processes, such as development, physiology, defense and stress responses, and adaptation to the changing environment [32]. The gene encoding RNase LC1—one of three cucurbits S-like RNase, which shows function in self-incompatibility—was the most downregulated by infection with ToLCNDV-ES. CsaV3_6G050450 was strongly induced in ToLCNDV-India-infected cucumber, and it encoded the NAC-domain protein (Appendix A). NAC proteins play a role as transcription factors in the promoter regions of different stress-related genes in response to biotic or abiotic stress [33].

### 2.4. GO Enrichment Analysis of Identified DEGs

To reveal the functional roles of selected DEGs in response to different ToLCNDV strains, GO enrichment analysis was performed to identify the GO terms of the DEGs shared between the two isolates. A total of 121 GO terms were identified in response to ToLCNDV-India, whereas only 18 GO terms were identified in the ToLCNDV infection group. Under the ToLCNDV-India condition, the number of GO terms (77 GO terms) for the upregulated genes was much higher than that for the downregulated genes (44 GO terms). For the ToLCNDV-ES condition, there were eight and ten GO terms for the up- and downregulated genes, respectively (Figure 6A). Next, the commonly identified GO terms among the four different groups (India-Up, India-Down, ES-Up, and ES-Down) were represented by a Venn diagram. Eight GO terms were commonly identified in the upregulated and downregulated gene groups in both strains. Only the fourth most common GO term was presented in ToLCNDV-India infection, whereas no GO term was found to be common between the up- and downregulated genes under the ToLCNDV-ES condition (Figure 6B). A total of 16 GO terms were identified between the ToLCNDV-ES and ToLCNDV-India infection groups (Figure 6C). The upregulated genes of both viral strains were dedicated to the cell wall (GO:0005618) and external encapsulating structure (GO:0030312) in the cellular component category. In the biological process group, genes involved in glucan metabolic process (GO:0044042), cell wall biogenesis (GO:0042546), cellular polysaccharide metabolic process (GO:0044264), cellular glucan metabolic progress (GO:0006073), cell wall organization or biogenesis (GO:0071554), oxidation–reduction process (GO:0055114), and polysaccharide metabolic process (GO:0005976) were highly upregulated, whereas the GO term for response to stress (GO:0006950) was enriched in the group of downregulated genes of both viral strains. The commonly identified GO terms assigned to molecular functions were terpene synthase activity (GO:0010333), carbon-oxygenase activity (GO:0016838), lyase activity (GO:0016829), dioxygenase activity (GO:0051213), and oxidoreductase activity (GO:0016701). In general, the number of genes assigned to the identified GO terms in ToLCNDV-India was much higher than that in ToLCNDV-ES.

### 2.5. Pathway Annotation of DEGs

The pathway enrichment in the four different groups (India-Up, India-Down, ES-Up, and ES-Down) showed that six pathways, including cellulose biosynthesis, suberin monomer biosynthesis, phenylpropanoid biosynthesis, galactolipid biosynthesis I, xyloglucan biosynthesis, and scopolin/esculin biosynthesis, were enriched in the upregulated genes, whereas eight pathways, consisting of flavonoid biosynthesis, UDP-D-xylose biosynthesis, L-cysteine degradation II, UDP-α-D-glucuronate biosynthesis, baicalein degradation, luteolintriglucuronide degradation, abscisic acid biosynthesis, and jasmonic acid synthesis, were enriched in the genes downregulated by ToLCNDV-India infection (Figure 7A). Further enrichment analysis of the up- and downregulated genes in the ToLCNDV-ES condition showed that the genes involved in flavonoid biosynthesis, pinobanksin biosynthesis, and flavonoid biosynthesis in equisetum were upregulated, whereas repressed genes were annotated in the abscisic acid biosynthesis, glyoxylate cycle, glyoxylate cycle and fatty acid degradation, baicalein degradation, luteolintriglucuronide degradation, and jasmonic acid synthesis pathways (Figure 7B). Both strains repressed the synthesis of defense hormones, such as jasmonic acid, and degradation of some antioxidants, such as luteolin and baicalin. However, there were some distinct responses of cucumbers to infection by the two isolates. ToLCNDV-India mostly induced genes related to the cell wall, such as suberin or cellulose, providing protection against pathogens, whereas ToLCNDV-ES upregulated the genes associated with flavonoids, which help regulate cellular activity and protect plants from oxidative stress factors.

### 2.6. Validation of the Transcriptome Data by qRT-PCR

To validate the DEGs, ten different genes were selected for qRT-PCR analysis, including two genes showing opposite expression (CsaV3_7G001540 and CsaV3_7G004500), three genes showing upregulation (CsaV3_6G038040, CsaV3_5G006290, and CsaV3_5G031640), and five genes showing downregulation (CsaV3_4G023890, CsaV3_2G017880, CsaV3_4G034400, CsaV3_3G014180, and CsaV3_3G022370) under both ToLCNDV with specific primer sets (Appendix A). The relative expression of these genes determined by qRT-PCR was correlated with the RNA-seq data (Figure 8). CsaV3_7G001540-encoding expansin-like A2 protein and an unknown function protein encoded by CsaV3_7G004500 were found to be induced in ToLCNDV-India and repressed in ToLCNDV-ES between both analysis methodologies. The upregulation and downregulation of other genes selected by qRT-PCR were consistent with the transcriptome results.

Simultaneously, the relative expression of the other nine candidate genes belonging to oxidation–reduction (CsaV3_7G010170, CsaV3_2G012080, CsaV3_4G023890, CsaV3_4G023920, and CsaV3_4G007760) and flavonoid pathways (CsaV3_2G036090, CsaV3_2G007940, CsaV3_3G027830, and CsaV3_4G027940) was confirmed using newly designed primers (Appendix A). The qRT-PCR data of five oxidation–reduction-related genes varied with the RNA-seq data (Figure 9A, Appendix A). All five genes were downregulated by more than 1.5 times in ToLCNDV-India infection compared with ToLCNDV-ES infection; however, the experimental data showed no difference in expression between the two conditions. For flavonoid pathway-related genes, the relative transcript accumulation obtained for CsaV3_2G036090 was most contrary with the RNA-seq data (Figure 9B, Appendix A). In ToLCNDV-ES infection, the plants increased the expression of this gene while reducing in the ToLCNDV-India condition, which differs from the RNA-seq data, which both showed upregulation.

## 3. Discussion

ToLCNDV is a major risk for many important crops worldwide. Classified into two main strains, the Asian and Mediterranean isolates have been studied and reported more recently. In the present study, the differences in symptom development in the ToLCNDV-India and ToLCNDV-ES infectious clones were found in cucumbers. The ToLCNDV-India isolated from Pakistan showed high pathogenicity with severe induced symptoms, whereas the ToLCNDV-ES clone from Italy showed no symptoms in infected plants. The examination of the viral titer at different time points revealed that ToLCNDV-ES showed late accumulation for both DNA A and DNA B (at 6–8 wpi) compared with ToLCNDV-India (at 3 wpi). This could be the reason for the accelerated disease phenotype variation between the two isolates. However, the ToLCNDV-ES-infected plants, which were observed up to 8 wpi, still exhibited a normal phenotype (data not shown), suggesting that late accumulation of the virus may not be the main factor affecting ToLCNDV pathogenicity in plants. To pinpoint the factors influencing disease phenotype variation in the same host plant, the swapped subgenomic DNA components were examined between the two ToLCNDV clones. The results showed that all plants infected with the swapped constructs showed similar leaf mosaic symptoms (Figure 4), and the severity of symptoms was lower than that of the original ToLCNDV-India-infected cucumber. The disease phenotype of the swapped genomic clones can complement the important role of the ToLCNDV DNA B component related to symptom development in plants, as shown in many previous studies [34,35]. Replacing the DNA B of ToLCNDV-ES with ToLCNDV-India DNA B resulted in the appearance of symptoms in infected plants compared to that in ToLCNDV-ES-infected plants. However, DNA A was also efficient in symptom development in the ToLCNDV-India isolate. Therefore, it is difficult to compare the functions of the different DNA components related to symptom development in the same host using the two ToLCNDV isolates. This difference may be due to the reaction of the plant defense system to various viruses, including distinct isolates of one virus. Because viral infection is a complex process involving an interaction between the host and virus, understanding host responses during different viral infections will help predict and compare the invasion strategies of the virus.

To explore the altered expression of cucumber genes after each ToLCNDV infection, RNA-seq was performed using high-throughput sequencing with total RNA from infected plants. The raw reads were mapped to cucumber “Chinese Long” on CuGenDB, and the significant DEGs were identified for the downstream analysis using the GO term and pathway enrichment. The number of identified common DEGs in the ToLCNDV-India condition was significantly higher (>10 times) than that in ToLCNDV-ES infection, demonstrating that ToLCNDV isolated from Pakistan had more severe effects on cucumbers. Functional enrichment analyses can help gain insight into the biological processes underlying tomato yellow leaf curl virus infection. In the ToLCNDV-India-infected samples, the over-represented GO categories included binding among upregulated genes and catalytic activity among downregulated genes. Plant viruses need essential steps to infect the plant, including cell-to-cell movement and viral replication and encapsidation and suppression of host defenses, and both are connected by means of cellular membranes and cell wall. The components of the cell wall and external encapsulating structure GO categories were found enriched in the DEGs in both infections. The DEGs annotated with these GO terms were upregulated, raising the possibility that ToLCNDV hampers the cell wall and damages its integrity to create favorable conditions to allow the virus to enter the cell and spread its infection. Moreover, biological processes, including oxidation–reduction and response to stress, were depressed in the two ToLCNDV infections. 

Pathway enrichment also provides features related to the host response to each ToLCNDV infection. Both infections reduced the expression of defense hormones and biosynthesis of oxidants in plants. Jasmonic acid is a hormone that plays an important role in host defense, including against geminivirus infections [36]. A few downregulated pathways were identified in one isolate, but were significant in other isolate. The flavonoid biosynthesis pathway was downregulated in ToLCNDV-India infection but was induced under ToLCNDV-ES conditions. Flavonoids are a large group of natural plant phenolics. They include almost 10,000 different compounds found in various plant species [37,38]. Flavonoids participate in many biological processes, such as the signaling pathway and responses to several abiotic stresses, and some flavonoids play important roles in the defense response against pathogens [39,40]. This means that ToLCNDV can be suppressed by flavonoids, and the virus reduces its expression. However, the host increased flavonoid synthesis, which may result in defense enhancement against ToLCNDV-ES and affect symptom development. Based on the GO term and pathway enrichment data, we focused on the DEGs related to oxidation–reduction and flavonoid pathways. Although the qRT-PCR results were different from the RNA-seq data, the flavonoid-related genes still showed some interesting results that can help to understand the mechanism of the two ToLCNDV isolates. Our hypothesis concentrates on the effect of ToLCNDV infection on flavonoid progression. In ToLCNDV-ES infection, the expression of genes related to flavonoid pathways was upregulated, which caused an increase in the concentrations of some types of flavonoids. Thus, the virus spread and viral proteins were limited, which led to low viral pathogenicity and weak induction of symptoms with unclear mechanisms. In contrast, ToLCNDV-India may be related to low amounts of flavonoids and lead to the development of symptoms in infected plants with full viral pathogenicity. In addition, previous reports have identified the QTL on chromosome 2 for the resistance of cucumber to ToLCNDV [30]. Thus, screening for significant DEGs on this chromosome was performed. 4-Coumarate-CoA ligase (4CL) (CsaV3_2G036090) is the gene with the highest induction in ToLCNDV-ES infection, whereas other genes that also encode 4CL were downregulated in the ToLCNDV-India condition. In plant defense reactions, 4CL plays a particularly important role, because it connects the phenylpropanoid pathway with the lignin and flavonoid branch pathways [41]. Some reports have revealed that altered expression of these genes delays or promotes tomato yellow leaf curl Sardinia virus infection, which may be related to the cellulose content inside plant cells [42]. Collectively, in ToLCNDV-India infection, the genes encoding 4CL were downregulated, similar to reports of geminivirus infection in cotton [43]. The relative expression of these genes was increased in ToLCNDV-ES but reduced in ToLCNDV-India infection by qRT-PCR data. This suggests that our assumption may be a good interpretation for the different phenotypes of cucumber under the two conditions in spite of more evidence needing to be clarified. Our results indicate the different infectivities of two distant geographical ToLCNDVs in cucumbers related to symptom development. In addition, we provide primary information based on the RNA-seq and confirmation by qRT-PCR for further analysis related to the different ToLCNDV infections.

## 4. Materials and Methods

### 4.1. Plant Growth Condition and Inoculation of Infectious Clones

In this study, the cucumber cultivar “Jellujon Baekchim”, which is similar to the known cultivar “Chinese long”, was used for all of the experiments. Cucumber seeds were obtained from a commercial company (NongWoo Bio, Suwon, Korea). Cucumber seeds were sown in the soil and kept in a plant growth chamber at 28/22 °C and 16/8 h light/dark periods. 

For the ToLCNDV infection, agrobacteria with empty pCAMBIA 1303 and two infectious clones of ToLCNDV-ES and ToLCNDV-India were prepared and inoculated to plants via pinpicking of the main apical shoot [44]. The presence of the virus in inoculated plants was verified at 21 dpi by PCR amplification with specific primer sets for each strain, according to previous studies [44]. Swapping of the DNA genomic components was conducted with two combinations, ToLCNDV-India A + ToLCNDV-ES B (marked as A_In_B_ES_) and ToLCNDV-ES A + ToLCNDV-India B (A_ES_B_In_), using a similar approach. Samples were collected at eight different time points (1, 2, 3, 4, 5, 6, 7, and 8 weeks postincubation) and were used to check the titer by quantitative PCR (qPCR) with previous methods [45].

### 4.2. Total RNA Extraction and Illumina Sequencing

For RNA-seq, four different plants infected with ToLCNDV-India or ToLCNDV-ES were used. In addition, four mock plants treated with agrobacteria-containing empty pCAMBIA 1303 were used as controls. Similar-sized young leaf samples from each group were harvested at 21 dpi and directly used for RNA extraction. Total RNA was extracted using the RNeasy Plant Mini Kit (Qiagen, Germany), according to the manufacturer’s instructions. The integrity of the total RNA was checked by 1% agarose gel electrophoresis, and the concentration was determined using an Epoch Microplate spectrophotometer (Biotek, Seoul, Korea). Twelve libraries were generated using the TruSeq RNA Library Prep Kit (version 2; Illumina, CA, USA), according to the manufacturer’s instructions. The generated mRNA libraries were paired-end (101 bp × 2) sequenced using a NovaSeq 6000 system (Macrogen). 

### 4.3. RNA Sequencing Data Analysis

For cucumber reference, the cucumber genome cultivar ”Chinese long” was used. The transcript sequences for the “Chinese long” version 3 were downloaded from the Cucurbits Genomics Database (CuGenDB) (http://cucurbitgenomics.org/ (accessed on 19 December 2022)). The ChineseLong_CDS_v3.fa.gz file was used to construct a reference database using the BWA index. The paired-end sequenced raw FASTQ files from individual libraries were mapped to the cucumber reference transcriptome using BWA-MEM with default parameters. The SAM files obtained were subjected to the eXpress program to calculate the abundance of mapped reads on the individual transcripts. 

### 4.4. Analysis of Differentially Expressed Genes 

The mapped read counts were used for the DEG analysis using DESeq2 in DEBrowser (version 1.22.4) [46]. Without filtering any low counts, all read counts were normalized using the MRN method. The ToLCNDV-India and ToLCNDV-ES conditions were compared with the mock condition. The DEGs were identified by DESeq2 with fit type (parametric), betPrior (False), test type (LRT), and shrinkage (None). Finally, the DEGs were selected using a fold change more than twice and adjusted the *p*-value to < 0.01 as a cutoff. Volcano plots were generated using log10-converted-adjusted *p*-values and log2-converted fold changes. Common DEGs of the two isolates were represented by Venn diagrams, displayed with DrawVenn, freely available at http://bioinformatics.psb.ugent.be/webtools/Venn/ (accessed on 10 March 2022).

### 4.5. Gene Ontology and Pathway Enrichment Analysis

The GO enrichment analyses were performed using the DEGs in four different groups, upregulated by ToLCNDV-India, upregulated by ToLCDNV-ES, downregulated by ToLCNDV-India, and downregulated by ToLCNDV-ES, against the cucumber cultivar “Chinese long” version 2 on CuGenDB, considering those with a *p*-value (Bonferroni) of 0.05 (cutoff *p*-value) as significantly enriched. 

Using the same CuGEnDB, the pathways annotated with DEGs were analyzed with a cutoff *p*-value of 0.05 based on the cucumber (“Chinese Long”) datasets.

### 4.6. Quantitative Real-Time Polymerase Chain Reaction

To validate the RNA-seq results, quantitative real-time PCR (qRT-PCR) was conducted with ten annotated genes in cucumber at 21 dpi. In addition, the qRT-PCR of nine selected candidate genes belonging to the oxidation–reduction process and flavonoid pathways were also processed using the leaf samples collected at 3 wpi.

For the reverse transcription, cDNA was synthesized from 1 µg of total RNA as a template using Oligo dT primers and Moloney murine leukemia virus (MMLV) reverse transcriptase (Bioneer, Daejeon, Korea). The qRT-PCR was performed using TB Green^®®^ Premix Ex Taq™ II (TliRNaseH Plus; TaKaRa Bio, Shiga, Japan) with primer sets (Appendix A). The PCR conditions using a Rotor-Gene Q thermocycler (Qiagen, Hilden, Germany) included 40 cycles—one cycle: 10 s denaturation at 95 °C, 15 s annealing at 60 °C, and 20 s polymerization at 72 °C. Ubiquitin was used for the internal normalization, and each reaction was replicated thrice. The data were analyzed using the 2^−ΔΔCt^ method [47]. The statistical analyses were performed using the GraphPad Prism software (GraphPad Software, San Diego, CA, USA).

## Figures and Tables

**Figure 1 ijms-24-02181-f001:**
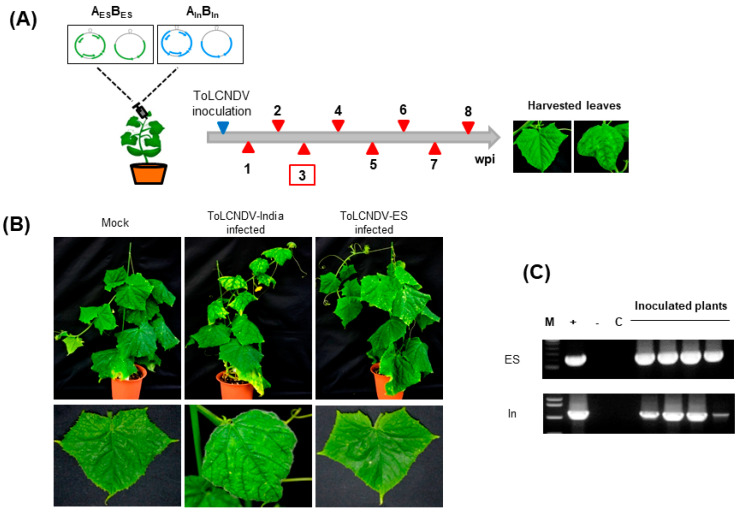
Infectivity of two ToLCNDV clones in cucumber: (**A**) Scheme of the experiment timeline from 1 to 8 weeks postinfection (wpi). At 3 wpi, symptoms appeared. and are marked with the red square. (**B**) Phenotype of the plants inoculated with empty Agrobacteria GV3101 (mock), ToLCNDV-India, and ToLCNDV-ES at 21 days postinfection (dpi). (**C**) PCR results to detect ToLCNDV in infected plants. Lane M, ladder; lane +, positive control; lane -, negative control; and lane C, mock plants.

**Figure 2 ijms-24-02181-f002:**
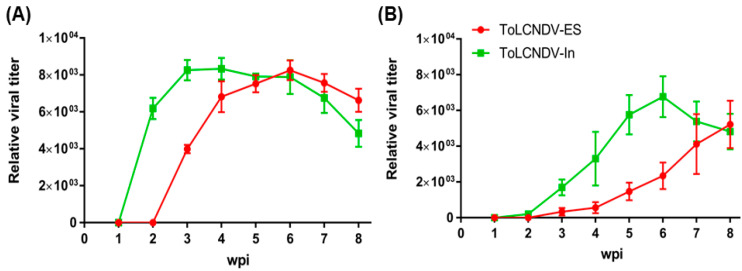
Relative ToLCNDV titer following the inoculated time in cucumber. Leaf samples were collected after 1–8 wpi, and relative amounts were measured by qPCR for the (**A**) DNA A and (**B**) DNA B components.

**Figure 3 ijms-24-02181-f003:**
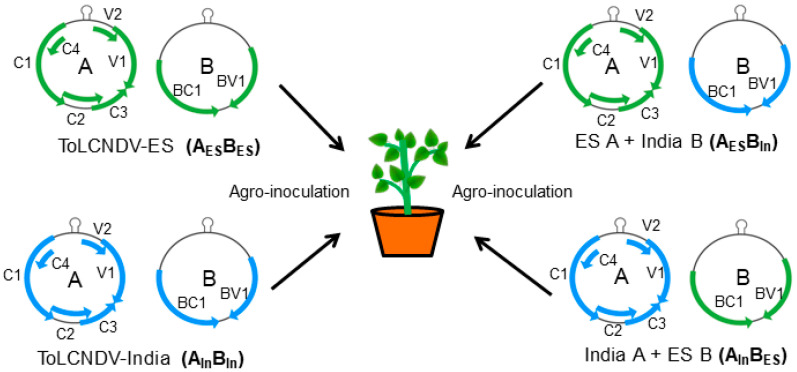
Experiment scheme for the swapped DNA components between two ToLCNDV clones. Component DNA A and B of each ToLCNDV were swapped and combined with other ToLCNDV components to inoculate the cucumber.

**Figure 4 ijms-24-02181-f004:**
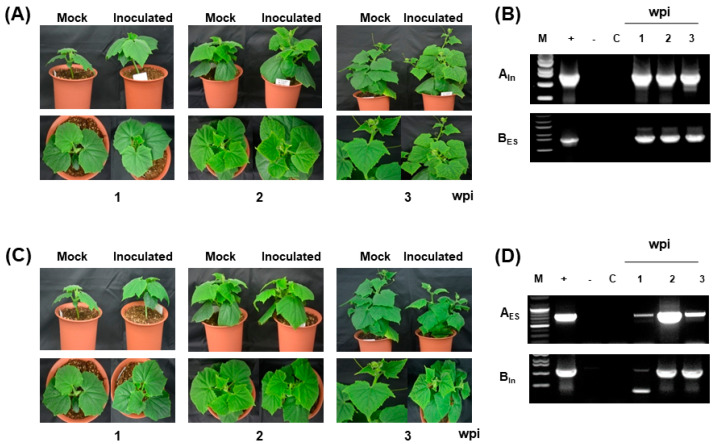
Summary of the swapped ToLCNDV component infection in cucumber: (**A**,**B**) phenotype and PCR results for ToLCNDV-India DNA A + ToLCNDV-ES DNA B (A_In_B_ES_); (**C**,**D**) infectivity of ToLCNDV-ES DNA A + ToLCNDV-India DNA B (A_ES_B_In_) by symptom appearance and PCR. Lane M, ladder; lane +, positive control; lane -, negative control; and lane C, mock plants.

**Figure 5 ijms-24-02181-f005:**
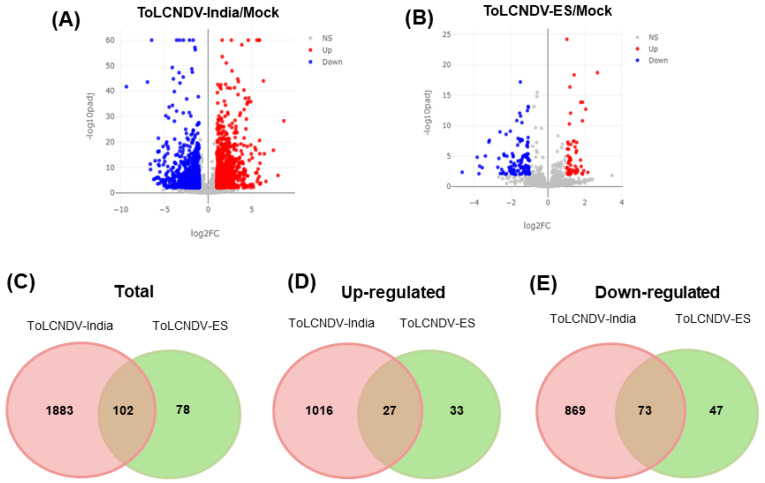
Identification of DEGs under each ToLCNDV isolate infection. Volcano plots were constructed with DEGs under (**A**) ToLCNDV-India and (**B**) ToLCNDV-ES infection. (**C**–**E**) Venn diagram representing the number of DEGs in total and the separately upregulated and downregulated genes, respectively.

**Figure 6 ijms-24-02181-f006:**
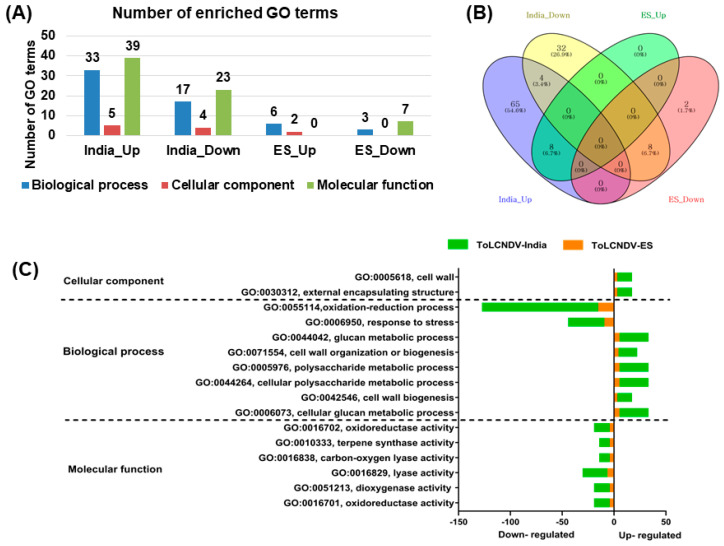
GO enrichment analysis under two ToLCNDV isolates infection. (**A**) The number of identified enriched GO terms. The graph shows the significant upregulated and downregulated GO terms involved in biological process (blue bars), cellular component (dark-red bars), and molecular function (green bars). (**B**) The Venn diagram represents the DEGs among 4 groups (India-Up, India-Down, ES-Up, and ES-Down). (**C**) The GO classification in both ToLCNDV conditions. The number of upregulated and downregulated genes calculated in ToLCNDV-India and -ES, represented by the green and orange colors according to the legend.

**Figure 7 ijms-24-02181-f007:**
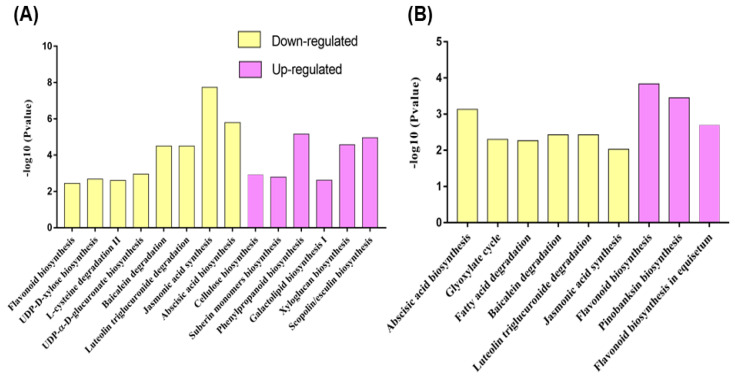
Pathway enrichment analysis of the upregulated and downregulated genes under ToLCNDV-India (**A**) and ToLCNDV-ES infection (**B**). Defense hormones, such as jasmonic acid and abscisic acid, were downregulated, while the genes related to the flavonoid pathway showed contrary expression under the two infections.

**Figure 8 ijms-24-02181-f008:**
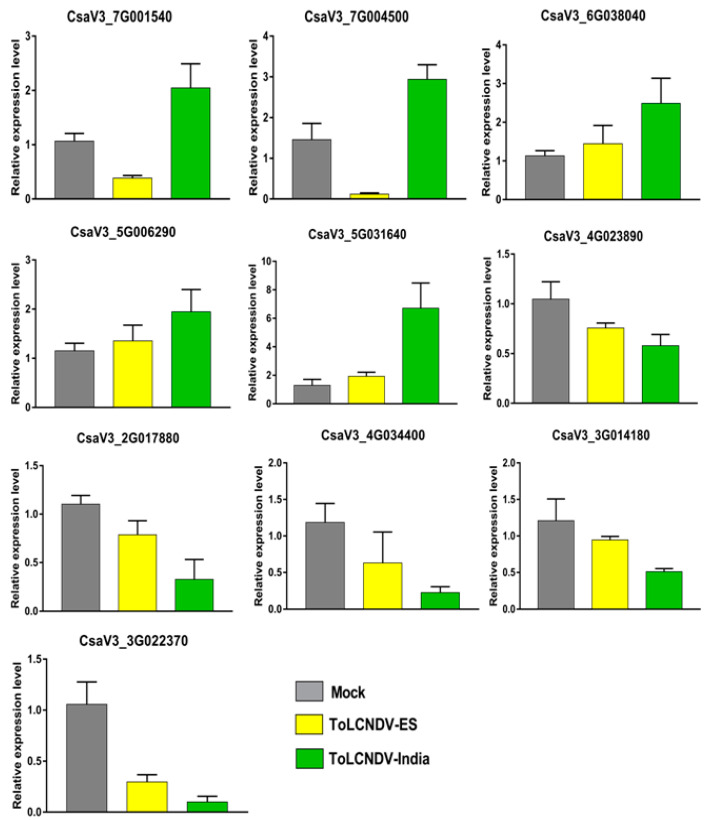
Quantitative RT-PCR to validate the RNA-seq data. Ten genes, including two genes that had opposite expressions, three upregulated genes, and five downregulated genes, were validated in the mock and infected plants.

**Figure 9 ijms-24-02181-f009:**
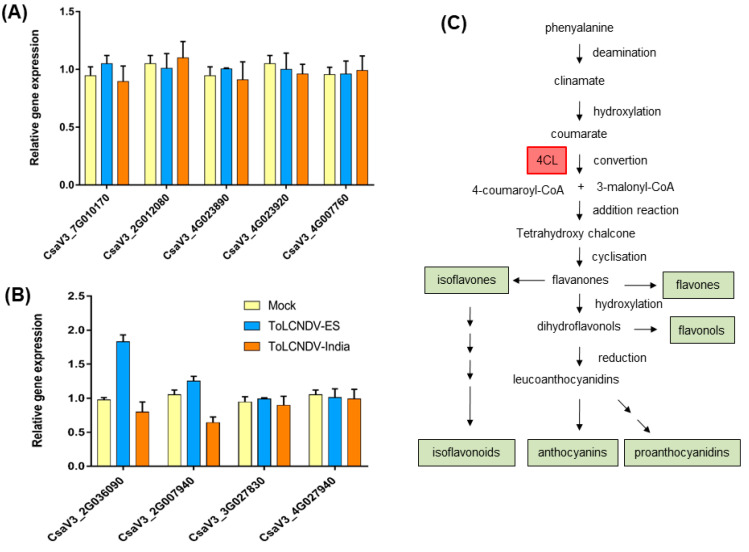
Relative expression of some potential genes in differently infected cucumber by qRT-PCR. Four genes belonging to the flavonoid pathway and five genes related to the oxidation/reduction process were chosen based on the RNA-seq data and their expression checked by qRT-PCR. The bar graphs show the changing expression of the candidate genes related to the (**A**) oxidation–reduction process and (**B**) flavonoid pathway at 21 dpi under the two ToLCNDV infections compared with the mock plants. (**C**) Flavonoid pathway with highlighted 4-coumarate-CoA ligase (4CL) in the red box.

**Table 1 ijms-24-02181-t001:** Infectivity of experimental ToLCNDV clones in *N. benthamiana* and cucumber (infectious clone, IC, of ToLCNDV-ES DNA A and B, A_ES_ and B_ES_, respectively, and of ToLCNDV-India DNA A and B, A_In_ and B_In_, respectively).

IC	*N. benthamiana*			Cucumber		
	Infectivity	Severity *	Symptoms	Infectivity	Severity *	Symptoms
Mock	0/9	-	-	0/7	-	-
A_ES_B_ES_	9/9	+++	Severe leaf curl, mosaic, stunting	5/7	-	No symptoms
A_In_B_In_	9/9	+++	Severe leaf curl, mosaic, stunting	7/7	++	Leaf curl, yellow mosaic
A_ES_B_In_	9/9	+++	Severe leaf curl, mosaic, stunting	4/7	+	Yellow mosaic on leaf
A_In_B_ES_	9/9	+++	Severe leaf curl, mosaic, stunting	3/7	+	Yellow mosaic on leaf

* Severity: -: infected but no symptom, +: infected and show 1 symptom, ++: infected and show 2 symptom, +++: infected and show 3 symptom.

**Table 2 ijms-24-02181-t002:** Summary of the raw data generated by sequencing of the three groups (mock, ToLCNDV-India, and ToLCNDV-ES).

Index	Infected Condition	Sample ID	Total Read Bases	Total Reads	GC (%)	Q20 (%)	Q30 (%)
1	ToLCNDV-ES	I-NDV-1-1	4,258,032,538	42,158,738	45.16	98.9	96.45
2	ToLCNDV-ES	I-NDV-1-2	4,264,281,004	42,220,604	45.62	98.95	96.54
3	ToLCNDV-ES	I-NDV-2-1	3,917,026,238	38,782,438	45.41	98.9	96.41
4	ToLCNDV-ES	I-NDV-2-2	3,668,776,722	36,324,522	45.09	98.98	96.54
5	Mock	Mock 1-1	3,895,794,422	38,572,222	45	98.89	96.38
6	Mock	Mock 1-2	3,596,973,600	35,613,600	44.75	98.89	96.34
7	Mock	Mock 2-1	3,797,215,190	37,596,190	44.95	98.84	96.22
8	Mock	Mock 2-2	3,931,060,996	38,921,396	45	98.88	96.29
9	ToLCNDV-India	Pa-NDV-1-1	3,523,436,914	34,885,514	44.79	97.54	92.9
10	ToLCNDV-India	Pa-NDV-1-2	4,123,268,238	40,824,438	44.95	97.74	93.41
11	ToLCNDV-India	Pa-NDV-2-1	4,111,628,796	40,709,196	44.98	97.58	93.02
12	ToLCNDV-India	Pa-NDV-2-2	3,201,276,608	31,695,808	44.97	97.42	92.7

## Data Availability

The raw dataset in this study is available in the Sequence Read Archive (SRA) repository with accession numbers SRR21227568, SRR21227569, and SRR21227570 in the BioProject PRJNA872216.

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
