# Peer review of "Transcriptional Analysis of the Differences between ToLCNDV-India and ToLCNDV-ES Leading to Contrary Symptom Development in Cucumber"

_ijms, 2023, doi:10.3390/ijms24032181_

Round 1

Reviewer 1 Report

 Fig. 2 7, 8 and 9 the writings mentioned in x-axis and z-axis is difficult to read, please chek this.

Please explain with more details the concetration of  RNA or cDNA used in the experiment.

Did the authors considering biochemical determination of antioxidant activity, flavonoids  to monitor the biosyntehsis of flavonoids after the gene expression

Please explain briefly the methdology of inoculaton of the vrus in the plants using agrobacterium

Reviewer 2 Report

Dear authors:

Thanks for submitting your manuscript!

Reading your manuscript, I understand that you explored the mechanisms underlying the different infectivity of two ToLCNDVs in cucumber through inoculation experiments followed by RNA-Seq, qRT-PCR, and downstream bioinformatics analysis. The paper was laid out in a clear manner, the methods adopted are appropriate, and the figures created are informative. I suggest the following revisions before launching it to a broader reader group.

  1. Though the language is clear overall, the manuscript suffers from many typos and grammar issues. It must be carefully proofread and polished. The concerns are:
    1. Missing spaces in many places: line 148, line 173, line 255, line 382, line 450, and many more.
    2. Grammar issues: line 146: should be "According to"; line 151: Should be "Component"; line 152: should remove "in"; line 161: "was" should be "were"; line 236: "with" should be "to"; lines 296-297; and many more.
    3. Unclear and confusing sentences: lines 130-131; line 333; lines 384-386.
  2. Line 338, what is TYLCV?
  3. Line 380, what is TYLCSV?
  4. In Figure 9, the qRT-PCR data does not match with RNA-Seq results. The potential causes and methods to solve the discrepancies must be discussed more.
  5. The authors hypothesized that in the ToLCNDV-ES-infected plants, there must be an increase in the concentrations of flavonoids, which limits the viral infection. I believe it would not be too complicated to experimentally quantify the flavonoids in both plants to provide more solid evidence to support the authors' hypothesis.
